# Yeast Diversity in the Qaidam Basin Desert in China with the Description of Five New Yeast Species

**DOI:** 10.3390/jof8080858

**Published:** 2022-08-16

**Authors:** Xu-Yang Wei, Hai-Yan Zhu, Liang Song, Ri-Peng Zhang, Ai-Hua Li, Qiu-Hong Niu, Xin-Zhan Liu, Feng-Yan Bai

**Affiliations:** 1College of Life Science and Agricultural Engineering, Nanyang Normal University, 1638 Wolong Road, Nanyang 473061, China; 2State Key Laboratory of Mycology, Institute of Microbiology, Chinese Academy of Sciences, Beijing 100101, China; 3College of Life Sciences, University of Chinese Academy of Sciences, No.19 (A) Yuquan Road, Shijingshan District, Beijing 100049, China; 4China General Microbiological Culture Collection Center, Institute of Microbiology, Chinese Academy of Sciences, Beijing 100101, China

**Keywords:** Qaidam basin, mars analog, yeast diversity, new species, halotolerant species

## Abstract

The Qaidam Basin is the highest and one of the largest and driest deserts on Earth. It is considered a mars analog area in China. In contrast to numerous studies concerning its geology, geophysical, and chemistry, relatively few studies have reported microbial diversity and distribution in this area. Here, we investigated culturable yeast diversity in the northeast Qaidam Basin. A total of 194 yeast strains were isolated, and 12 genera and 21 species were identified, among which 19 were basidiomycetous yeasts. *Naganishia albida*, *N. adeliensis*, and *Filobasidium magnum* were the three most dominant species and were distributed in thirteen samples from eight locations. Five new species (*Filobasidium chaidanensis*, *Kondoa globosum*, *Symmetrospora salmoneus*, *Teunia nitrariae*, and *Vishniacozyma pseudodimennae*) were found and described based on ITS and D1D2 gene loci together with phenotypic characteristics and physiochemical analysis. Representative strains from each species were chosen for the salt-tolerant test, in which species showed different responses to different levels of NaCl concentrations. Further, the strain from soil can adapt well to the higher salt stress compared to those from plants or lichens. Our study represents the first report of the yeast diversity in the Qaidam Basin, including five new species, and also provides further information on the halotolerance of yeasts from the saline environment in mars analog.

## 1. Introduction

Mars analogs are places on Earth that possess one or more geological, geochemical, and geomorphic features similar to those found on Mars, either current or past [1,2]. The possibility of the emergence of life on the early Mars stage has been shown due to the presence of liquid water, carbon, electron donors, and acceptors that can drive microbial metabolism [3,4,5,6,7,8]. Due to the expensive and long-term dependent organization of space exploration, terrestrial analogs can serve as models for studies on the habitability of Mars, which has been a priority of Mars science investigations [5]. Generally, the terrestrial analogs on Earth are characterized by extreme environmental conditions, such as hyper-arid desert, saline or hypersaline habitat, thermal acidic spring, and cryogenic and dry Antarctica [9,10,11,12,13,14,15]. The Atacama Desert in Chile, the Mars Desert Research Station in Utah, USA, and Rio Tinto in Huelva, Southwestern Spain are among the famous and well-studied Mars analog environments. Special issues of *International Journal of Astrobiology* (‘Astrobiology field research in Moon/Mars analog environments’ in 2011), *Life* (‘Planetary Exploration: Habitats and Terrestrial Analog’ in 2014), *Antonie van Leeuwenhoek* (‘Microbiology of the Atacama Desert’ in 2018), and *Environmental Microbiology* (‘Ecophysiology of Extremophiles’ in 2021) have been published for systematical studies on the Mars analog environments on Earth.

Microbial composition and ecology in different Mars analogs have drawn attention due to their ecological and astrobiological importance. Microbiological studies in Mars analogs performed in the past decades have contributed to our better understanding of the strategy and spatiotemporal limit of life on Earth [16,17,18,19,20,21,22,23]. However, these studies mostly focused on bacteria [18,20,24,25], and the diversity and ecology of fungi in Mars analogs remain largely unknown.

Yeasts are unicellular or dimorphic fungi with a stable free-living unicellular state during their life cycles [26]. Yeasts are considered ubiquitous and have been found in highly diversified substrates and environments, including extremely dry or cold habitats [27]. An increasing number of studies have shown that yeasts can inhabit the Antarctic, glacier, acidic, hypersaline, hyper-arid, and desert areas [28,29,30,31,32,33,34]. Nevertheless, the occurrence and diversity of yeasts in extreme environments have not been fully investigated.

The Qaidam Basin is located in the North Tibetan Plateau, China, and is the highest desert in terms of altitude and one of the largest and driest deserts on Earth. It has an average elevation of ~2800 m and is considered a terrestrial analog site of Mars [12]. This area harbors Mars-like extreme environments, e.g., hyper-arid, salt deposits, and high-UV, and similar mineral compositions and geomorphological structures to terrains on Mars [12,35]. The geology, geophysiology, and chemistry of the Qaidam Basin have been well characterized [36,37,38], and microbial diversities focusing on bacteria and archaea in different parts of this area using culture-independent methods have been performed in recent years [39,40,41]. However, the fungal diversity in the Qaidam Basin remains to be revealed.

In this study, we collected soil, soil crust, plant, and lichen samples in the northeast Qaidam Basin in Qinghai province and investigated the occurrence and diversity of yeasts in these samples using a culture-based approach. A total of 21 species, including five new species, were identified from the isolated yeast strains. In addition, the salt-tolerant ability of the representative yeast strains isolated was tested to evaluate their biotechnological potential. To our knowledge, this is the first report of the diversity of culturable yeasts within the Qaidam Basin. The aim of this study is to characterize the yeast community inhabiting this specific environment and identify potential salt-tolerant yeasts that could be of importance for astrobiology and biotechnology.

## 2. Materials and Methods

### 2.1. Sampling Sites

A total of 27 samples, including 19 surface soil and soil crust samples and eight whole plant tissue and lichen samples, were selected from Delingha city in the Qaidam Basin located in Qinghai province in April 2021 (Figure 1 and Table 1). Samples were sealed in sterilized plastic bags and transported to the laboratory at ambient temperatures.

### 2.2. Soil Chemical Analyses

The soil was suspended 1:5 (*w*/*v*) in sterile water and shaken at 180 rpm at 20 °C for 30 min. Soil pH and salinity were determined using the HQ40d Portable Meter (Hach Company, Loveland, CO, USA) (Table 1).

### 2.3. Yeast Isolation

All samples were processed immediately after returning to the laboratory. Approximately 2 g of soil, soil crust, and fragmented plant and lichen tissue samples were suspended in 10 mL of sterile water and shaken at 180 rpm for 24 h. Aliquots of 0.2 mL of each suspension were spread on potato dextrose agar (PDA, *w*/*v*, 20% potato infusion, 2% glucose, 2% agar), 1/2 PDA, and 1/10 PDA plates supplemented with 500 mg L^−1^ of ampicillin and 500 mg L^−1^ of streptomycin sulfate. The plates were incubated at 17 °C and 25 °C, respectively, for 3–7 d. All the yeast colony morphotypes were isolated, purified, and preserved in 20% glycerol at −80 °C for further study.

### 2.4. Molecular Phylogenetic Analysis

Genomic DNA was extracted from yeast cells that were actively growing on a YPD medium (*w*/*v*, 1% yeast extract, 2% peptone, 2% glucose, 2% agar) following the protocol described by Makimura et al. [42]. The ITS region (including the 5.8 S rRNA gene) and D1/D2 domain of the LSU rRNA gene were amplified and sequenced using the primers ITS1 and NL4 as described previously [43].

Sequences were inspected and assembled using the SeqMan program in the Lasergene 7 software package (DNASTAR Inc., Madison, WI, USA). They were then aligned using MAFFT version 7 using the G-INS-I algorithm [44]. The alignment was manually edited and trimmed, and minor gaps in all alignments were deleted. Maximum likelihood (ML) and Bayesian analyses were performed for separate and combined gene sequences using RAxML-HPC V.8 [45] and MrBayes 3.2.1 [46] on the CIPRES Science Gateway (http://www.phylo.org (accessed on 23 January 2022)), respectively. ML analysis was implemented with the rapid bootstrap algorithm with 1000 replicates and a subsequent search for the best-scoring tree using the GTRGAMMA model. The best-fit evolution model of separate and combined nucleotide sequences was determined using the Bayesian Information Criterion (BIC) jModeltest [47]. Five million generations were run with four Markov chains, sampling every 5000 generations.

Branches that received bootstrap values (BP) for Maximum likelihood and Bayesian posterior probabilities (BPP) greater than or equal to 70% and 0.95, respectively, were considered as significantly supported. The GenBank accession numbers for the sequences of the ITS region, LSU rRNA gene D1/D2 domain used in this study are listed in Appendix A. The alignments and trees were deposited in TreeBase (www.treebase.org (accessed on 1 March 2022), No. 29479).

### 2.5. Phenotypical Characterization

The phenotypic and physiological characters were examined according to standard methods used in yeast taxonomy [48]. The potential sexual cycles of all new species were investigated using CMA (*w*/*v*, 2.5% corn meal infusion, 2% agar), PDA, V8 (*w*/*v*, 10% V8 juice, 2% agar), and YCB (*w*/*v*, 1.17% yeast carbon base, 2% agar). A loopful of cells of each test strain was incubated singly or mixed on an agar plate and incubated at 17 °C for one month. The ballistoconidium-forming activity of all new species was observed by the inverted-plate method [49] on CMA at 17 °C. After 3 to 14 d, the glass slide containing the discharged spores was removed for examination under the microscope.

### 2.6. Salinity Tolerance Analysis

Salinity tolerance tests were performed on the representative strains of each species using spot plates. All the growth experiments were conducted at 20 °C in YPD agar plates supplemented with 0%, 3%, 6%, 9%, 12%, and 15% (*w*/*v*) NaCl, respectively. Briefly, strains were precultured in 3 mL YPD broth for 48 h at 20 °C with shaking at 180 rpm. Optical density (OD) at 600 was measured, and the cell suspension was adjusted to OD 0.8 before a ten-fold dilution series was carried out. An aliquot of 4 μL of each dilution of cell suspension was then spotted onto YPD agar plates supplemented with different concentrations of NaCl and incubated for five days at 20 °C.

### 2.7. Statistical Analysis

Standard statistical analyses were performed in the R project (v3.3.1) [50]. The statistical significance was calculated using the ANOVA or Kruskal–Wallis test executed by the *aov* function and *kruskal.test* function of R package multcomp and FSA, respectively.

## 3. Results

### 3.1. Yeast Diversity

Yeast strains were isolated from 10 of the 19 soil samples and six of the eight plant and lichen samples collected from Delingha city in the Qaidam Basin. The soil samples with more than 34.32‰ salinity and the soil crust samples were yeast negative (Table 1). A total of 194 yeast strains were isolated, and the abundance of yeasts varied depending on the substrate, incubation temperature, and medium (Figure 2). A total of 153 and 41 yeast strains were isolated from the plant and lichen tissues and soil samples, respectively. The PDA, 1/2 PDA, and 1/10 PDA agar produced 95, 68, and 31 yeast strains, respectively. A total of 91 and 103 yeast strains were isolated from the plates incubated at 17 °C and 25 °C, respectively (Appendix A). Comparative analysis indicated that both the numbers of yeast strains and species from plant and lichen tissue samples were significantly higher than those from soil samples, whereas no significant difference in the abundance and species number was observed regarding the temperature and medium used for yeast isolation (Figure 2). In addition to the shared species, different substrates, temperatures, and media harbored their own yeast species (Figure 2).

The sequences of the D1/D2 domain of the 194 isolated strains were determined. The closest relative of each strain was recognized based on a blast search against the GenBank database using the BLASTn tool. The strains were classified to the species level based on the threshold of >99% sequence identity with the type strain of a described species in the ITS region or D1/D2 domain [51,52,53]. A total of 21 species belonging to 12 genera were identified from the isolated strains. Among the species identified, 19 species were basidiomycetous yeasts, including *Dioszegia hungarica*, *Filobasidium magnum*, *Kondoa sorbi*, *Naganishia adeliensis*, *N. albida*, *Papiliotrema laurentii*, *Teunia korlaensis*, *Udeniomyces puniceus*, *Vishniacozyma dimennae*, *V. tephrensis*, *V. victoriae,* and eight undescribed species, i.e., *Fibulobasidium* sp., *Filobasidium* sp., *Kondoa* sp., *Symmetrospora* sp., *Teunia* sp., and *Vishniacozyma* sp. (Appendix A), and two were ascomycetous yeast species, namely *Diutina catenulata* and *Pichia kudriavzevii*. The most abundant yeast genus was *Naganishia*, followed by *Filobasidium* and *Teunia*. The *Naganishia* species, viz. *N. albida* and *N. adeliensis*, accounted for 39.2% of the total number of isolates. To a lesser extent was the genus *Filobasidium*, of which an undescribed species, *Filobasidium* sp. formed the majority of the isolates. The strains belonging to the genus *Teunia* accounted for 9.8% of the total yeast strains isolated, and half belonged to an undescribed species of the genus. On the whole, 24.7% of the isolated strains belonged to undescribed species, and most of these were isolated from plant samples (Appendix A). Among the 21 species identified, three species, i.e., *Filobasidium magnum*, *Naganishia adeliensis,* and *N. albida*, were the most dominant species, which occurred in seven to eight samples collected from different locations, while thirteen species, i.e., *Dioszegia hungarica*, *Fibulobasidium* sp., *Kondoa sorbi*, *Kondoa globosum* sp. nov., *Kondoa* sp. *Teunia nitrariae* sp. nov., *Vishniacozyma dimennae*, *Vishniacozyma pseudodimennae* sp. nov., and *Vishniacozyma tephrensis*, occurred only in one sample or location (Appendix A).

### 3.2. Salt-Tolerant Ability

Representative strains from lichen, plant, and soil samples were subjected to salt-tolerant ability tests using spot plate assay. Different species had different responses to concentrations of NaCl. In detail, all tested strains of 21 species identified in this study could grow well with NaCl concentrations up to 6% (Appendix A), while no strains could grow on the plate with NaCl concentrations of 15% (data not shown). It suggested that their growth was apparently prevented when NaCl concentration increased to 15%. Four species, such as *Dioszegia hugarica*, *Fibulobasidium* sp., *Kondoa globosum*, and *Teunia korlaensis,* were particularly sensitive to 9% NaCl concentration, while the other species, especially *Diutina catenulata*, *Filobasidium magnum*, *Kondoa sorbi*, *Naganishia adeliensis, Papiliotrema laurenti*, *Pichia kudriavzevii*, *Symmetrospora salmoneus*, and *Vishniacozyma victoriae*, showed to be unaffected by this level of salt exposure (Appendix A). Several strains, such as *F. chaidanensis*, *Kondoa* sp. *N. albida*, *Teunia nitrariae, Teunia* sp., *Udeniomyces puniceus*, *Vishniacozyma dimennae*, *V. pseudodimennae*, and *V. tephrensis,* showed some ability to grow at 9% NaCl concentration. In contrast, most strains appeared to be sensitive to exposure to NaCl concentration of 12%. The strains of *F. magnum*, *P. laurenti*, and *V. victoriae* seemed to be relatively unaffected by this level of NaCl exposure (Appendix A). These three species exhibited a similar tendency, that is, no difference, in all tested strains in growth ability regardless of their origin from plant and soil samples with different NaCl concentrations, except for the strain of *F. magnum* from lichen (Figure 3). The strains of *N. adeliensis* and *S. salmoneus* from the soil sample showed better growth than those from plant or lichen thallus at 12% NaCl concentration, indicating that the strain from soil samples can adapt well to the higher salt stress.

### 3.3. Phylogeny and Taxonomy of Novel Yeast Species

#### 3.3.1. Phylogeny

Among the yeast strains isolated, 43 strains were unable to be identified as any species because of their significant difference from any described yeast species in the sequence of the D1/D2 domain. The ITS sequences of these strains were then determined to confirm their novelty and phylogenetic positions.

A total of 32 strains, represented by CGMCC 2.6791, CGMCC 2.6792, CGMCC 2.6793, CGMCC 2.6794, CGMCC 2.6795, CGMCC 2.6796^T^, 25-0-5-256-2, and 25-0-5-256-6 isolated from plant tissues formed a separate clade in the genus *Filobasidium* in the tree obtained from the combined ITS and D1/D2 sequences (Figure 4A). These strains possessed identical ITS and D1/D2 sequences or differed by only one nucleotide in the two regions and thus were conspecific. They differed from the type strain of closely related species *F. oeirensis*, *F. floriforme*, *F. elegans*, and *F. magnus* by seven to nine nt (~1.1–1.5%) and 26–36 nt (~4.1–5.8%) mismatches in the D1/D2 (Appendix A) and ITS regions, respectively. Therefore, these strains represent a new species in *Filobasidium*, for which the name *F. chaidanensis* sp. nov. was proposed.

Strain CGMCC 2.6805^T^ was located in the *Kondoa* clade and closely related with *K. sorbi* CGMCC 2.2303^T^ in the tree obtained from the combined ITS and D1D2 sequences (Figure 4B). The two strains differed from each other by 9 nt (~1.4%) and 55 nt (~8.4%) mismatches in the D1/D2 and ITS regions, respectively. An undescribed strain ‘*Bensingtonia*’ sp. CCFEE 5424 isolated from rock samples [54] and two unpublished strains ‘Pucciniomycotina’ sp. RP185 (AB727159) and RP186 (AB727160) isolated from plant materials in Japan showed close relationships with strain CGMCC 2.6805^T^ (Appendix A), but they differed from the latter by more than 1% bases in the D1/D2 domain. The result suggests that strain CGMCC 2.6805^T^ represents a novel species in the genus *Kondoa*, for which the name *K. globosum* sp. nov. was proposed.

Three strains, CGMCC 2.6800, CGMCC 2.6801^T^, and 17-256-4 isolated from sandy soil and plant tissues possessed similar sequences with no more than one nt difference in the ITS and D1D2 regions, indicating they are conspecific. They were clustered in the *Symmetrospora* clade with strong support in the tree constructed from the combined ITS and D1/D2 sequences (Figure 4C). *S. foliicola* CBS 8075^T^ is closely related to these three strains but without significant support (Figure 4C and Appendix A). Ten to eleven nt (~1.6–1.7%) and eighteen nt (~3.1%) differences were observed in the ITS region and D1D2 domain, respectively, between these three strains and *S. foliicola*. *S. rhododendri*, *S. coprosmae*, and *S. oryzicola* were located as a sister group to the group represented by CGMCC 2.6801^T^ and *S. follicola* but differed from the group CGMCC 2.6801^T^ by 7–10 nt (~1.3–1.6%) mismatches in the D1D2 domains and more than 7.2% (44–51 bp) mismatches in the ITS regions. Thus, these three strains represent a novel *Symmetrospora* species, for which the name *S. salmoneus* sp. nov. was proposed.

Six strains, represented by strain CGMCC 2.6797^T^, formed a well-supported clade (Figure 4D). Four nt and one nt mismatches in the D1D2 and ITS regions, respectively, were found among these six strains indicating they were conspecific. The closest relative of the CGMCC 2.6797^T^ group is *Teunia korlaensis*, but differed from the type strain of the latter by eight to nine nt (~1.3–1.5%) in the D1D2 domains and 22 nt (~3.9%) in the ITS regions, respectively. They also had a close relationship with *T. rosae* in the D1/D2 tree (Appendix A). They differed from the type strain of the latter by three to five nt (~0.5–0.8%) in the D1D2 domains but 38 to 40 nt (~6.8–7.1%) mismatches in the ITS regions, respectively. Therefore, a novel species, *Teunia nitrariae*, was proposed to accommodate these six strains.

Strain CGMCC 2.6790^T^ was located in the genus *Vishniacozyma*. It was closely related to *Vishniacozyma dimennae* (Figure 4E and Appendix A) but differed from the type strain of *V. dimennae* by ten nt (~1.6%) and 19 nt (~4.0%) mismatches in the D1D2 domains and ITS regions, respectively, suggesting that CGMCC 2.6790^T^ represents a new species in *Vishniacozyma*, for which the name *V. pseudodimennae* sp. nov. is proposed.

#### 3.3.2. Taxonomy

***Filobasidium chaidanensis*** X.Z. Liu, F.Y Bai, and X.Y. Wei, **sp. nov.** (Figure 5A)

MycoBank: MB 842980

Etymology: the specific epithet *chaidanensis* refers to the geographic origin of the type strain, Chaidan county, Qinghai province.

Culture characteristics: After growth on YM agar for 7 days at 20 °C, cells were globose or subglobose, 3.3–6.6 μm × 2.8–6.7 μm, and occurred singly or in pairs. Budding was polar. After 7 days of growth on YPD agar at 20 °C, the streak culture was whitish to pale greyish-cream, smooth, and butyrous. The margin was entire. In Dalmau plated culture on corn meal agar, pseudohyphae and hyphae were not formed. Sexual structures were not observed on YCB, 5% MEA, PDA, V8, and CMA agar. Ballistoconidia were not produced on corn meal agar.

Physiological and biochemical characteristics: Glucose was not fermented. Glucose, galactose (positive or weak), L-sorbose (weak or slow), sucrose, maltose, cellobiose (positive or delayed), trehalose, lactose, melibiose (weak), raffinose (positive or delayed), melezitose, inulin (weak), soluble starch, D-xylose, L-arabinose, D-arabinose (weak), D-ribose (weak), L-rhamnose, methanol (weak), ethanol (positive or delayed), glycerol (positive or delayed or weak or delayed and weak), ribitol (weak or delayed weak), galactitol (weak), D-mannitol, D-glucitol, Methyl-α-D-glucoside (positive or delayed), salicin, D-gluconate (weak), DL-lactate (weak), succinate, citrate (positive or weak), myo-inositol (weak) and xylitol (positive or delayed) were assimilated as sole carbon sources. D-glucosamine, erythritol, hexadecane, and N-Acetyl-D-glucosamine were not assimilated as sole carbon sources. Ammonium sulfate, potassium nitrate, cadaverine, L-lysine, and ethylamine (variable) were assimilated as sole nitrogen sources. Sodium nitrite was not assimilated as the sole nitrogen source. Growth in the vitamin-free medium was positive. Starch-like compounds were produced. Urease activity was positive. Diazonium Blue B reaction was positive. The maximum growth temperature was 29 °C.

Physiologically, *F. chaidanensis* differed from its close relatives *F. oeirensis*, *F. floriforme*, *F. elegans*, and *F. magnus* in the ability to assimilate inulin, methanol, and DL-lactate and its ability to grow in a vitamin-free medium.

Typus: China, Chaidan town, Delingha City, Qinghai province, obtained from undescribed plant tissue, April 2021, X.L. Wei, holotype CGMCC 2.6796 preserved in a metabolically inactive state. Ex-type culture has been deposited in the Japan Collection of Microorganisms (JCM) as JCM 35,461 (=25-P263-B).

***Kondoa globosum*** X.Z. Liu, F.Y Bai, and X.Y. Wei, **sp. nov.** (Figure 5B)

MycoBank: MB 842983

Etymology: the specific epithet *globosum* refers to the globose vegetative cells of the type strain.

Culture characteristics: After growth on YM agar for 7 days at 20 °C, cells were subglobose to globose, 4.5–7.6 μm × 4.3–7.3 μm, and occurred singly or in pairs. Budding was polar. After 7 days of growth on YPD agar at 20 °C, the streak culture was cream-colored, smooth, and dull. The margin was entire. In Dalmau plated culture on corn meal agar, pseudohyphae were not formed. Hyphae were seldom formed. Sexual structures were not observed on YCB, 5% MEA, PDA, V8, and CMA agar. Ballistoconidia were not produced on corn meal agar.

Physiological and biochemical characteristics: Glucose was not fermented. Glucose, maltose, cellobiose (slow), trehalose, lactose (weak), melibiose (weak), soluble starch (delayed weak), glycerol (slow), ribitol (slow), D-mannitol (weak), salicin (weak), DL-lactate (weak), and succinate (delayed) were assimilated as sole carbon sources. Galactose, L-sorbose, sucrose, raffinose, melezitose, inulin, D-xylose, L-arabinose, D-arabinose, D-ribose, L-rhamnose, D-glucosamine, methanol, ethanol, erythritol, galactitol, D-glucitol, Methyl-α-D-glucoside, D-gluconate, citrate, myo-inositol, xylitol, hexadecane, and N-Acetyl-D-glucosamine were not assimilated as sole carbon sources. Ammonium sulfate, potassium nitrate, cadaverine, L-lysine, and ethylamine were assimilated as sole nitrogen sources. Sodium nitrite was not assimilated as the sole nitrogen source. Growth in the vitamin-free medium was positive. Starch-like compounds were not produced. Urease activity was positive. Diazonium Blue B reaction was positive. The maximum growth temperature was 28 °C.

Physiologically, *K. globosum* differed from its close relatives *K. sorbi* in its ability to assimilate potassium nitrate and ethylamine and inability to assimilate raffinose, D-xylose, L-arabinose, D-glucitol, and D-gluconate.

Typus: China, Chaidan town, Delingha City, Qinghai province, obtained from soil, April 2021, X.L. Wei, holotype CGMCC 2.6805 preserved in a metabolically inactive state. Ex-type culture has been deposited in the Japan Collection of Microorganisms (JCM) as JCM 35,467 (=17-P263-8-2).

***Symmetrospora salmoneus*** X.Z. Liu, F.Y Bai, and X.Y. Wei, **sp. nov.** (Figure 5C)

MycoBank: MB842985

Etymology: The specific epithet refers to the salmon color of the colony of the strains in this species.

Culture characteristics: After growth on YM agar for 7 days at 20 °C, cells were globose and subglobose, 4.1–6.6 μm × 3.6–6.3 μm, and occurred singly or in pairs. Budding was polar. After 7 days of growth on YPD agar at 20 °C, the streak culture was salmon-pink or orange-colored, smooth, slightly wrinkled, butyrous, and semi-glossy. The margin was entire. In Dalmau plated culture on corn meal agar, pseudohyphae and hyphae were not formed. Hyphae were seldom formed when mixed strains were cultured (Figure 5D). Sexual structures were not observed on YCB, 5% MEA, PDA, V8, and CMA agar. Ballistoconidia were not produced on corn meal agar.

Physiological and biochemical characteristics: Glucose is not fermented. Glucose, galactose (weak), sucrose (weak), maltose (weak), melibiose (weak), soluble starch (weak), D-xylose (weak), L-arabinose (weak), D-ribose (delayed or delayed weak), ethanol (delayed weak), glycerol (delayed or weak), ribitol (weak), D-mannitol (variable), D-glucitol (weak), salicin (delayed weak), DL-lactate (weak), succinate (variable), and xylitol (delayed weak) are assimilated as sole carbon sources. L-sorbose, cellobiose, trehalose, lactose, raffinose, melezitose, inulin, D-arabinose, L-rhamnose, D-glucosamine, methanol, erythritol, galactitol, Methyl-α-D-glucoside, D-gluconate, citrate, myo-inositol, hexadecane, and N-Acetyl-D-glucosamine are not assimilated as sole carbon sources. Ammonium sulfate, potassium nitrate (variable), cadaverine (variable), and ethylamine (variable) are assimilated as sole nitrogen sources. Sodium nitrite and L-lysine are not assimilated as sole nitrogen sources. Growth in the vitamin-free medium is positive (weak). Starch-like compounds are not produced. Urease activity is positive. Diazonium Blue B reaction is positive. The maximum growth temperature is 29 °C.

Physiologically, *S. salmoneus* differed from its close relatives *S. foliicola* in the inability to assimilate L-sorbose, cellobiose, trehalose, melezitose, D-arabinose, D-gluconate, and sodium nitrite.

Typus: China, Chaidan town, Delingha City, Qinghai province, isolated from sand, April 2021, X.L. Wei, holotype CGMCC 2.6801 preserved in a metabolically inactive state. Ex-type culture has been deposited in the Japan Collection of Microorganisms (JCM) as JCM 35,466 (=17-P263-5-2).

***Teunia nitrariae*** X.Z. Liu, F.Y Bai, and X.Y. Wei, **sp. nov.** (Figure 5E)

MycoBank: MB 842986

Etymology: The specific epithet refers to *Nitraria*, the plant genus from which the type strain was isolated.

Culture characteristics: After growth on YM agar for 7 days at 20 °C, cells were subglobose, globose to ovoid, 3.2–6.4 μm × 2.5–5.7 μm, and occurred singly or in pairs. Budding was polar. After 7 days of growth on YPD agar at 20 °C, the streak culture was yellowish-cream, smooth, and glossy. The margin was entire. In Dalmau plated culture on corn meal agar, pseudohyphae and hyphae were not formed. Sexual structures were not observed on YCB, 5% MEA, PDA, V8, and CMA agar. Ballistoconidia were not produced on corn meal agar.

Physiological and biochemical characteristics: Glucose was not fermented. Glucose, galactose, L-sorbose (variable), maltose (weak), cellobiose, trehalose (positive or weak), lactose (positive or slow), melibiose (weak), soluble starch (variable), D-xylose, L-arabinose, D-arabinose (variable), D-ribose (positive or weak), L-rhamnose (positive or weak), ethanol (delayed weak), glycerol (weak or delayed weak), ribitol (weak or delayed weak), galactitol (delayed), D-mannitol, D-glucitol, salicin (positive or weak), D-gluconate (positive or delayed or delayed weak), DL-lactate (delayed weak), succinate (delayed), myo-inositol (delayed weak), and xylitol (positive or delayed) were assimilated as sole carbon sources. Sucrose, raffinose, melezitose, inulin, methanol, Methyl-α-D-glucoside, citrate, D-glucosamine, erythritol, hexadecane, and N-Acetyl-D-glucosamine were not assimilated as sole carbon sources. Ammonium sulfate, potassium nitrate (variable), cadaverine, L-lysine, and ethylamine (variable) were assimilated as sole nitrogen sources. Sodium nitrite was not assimilated as the sole nitrogen source. Growth in the vitamin-free medium was positive. Starch-like compounds were produced (variable). Urease activity was positive. Diazonium Blue B reaction was positive. The maximum growth temperature was 29 °C.

Physiologically, *T. nitrariae* differed from its close relatives, *T*. *korlaensis* and *T*. *rosae,* in the ability to assimilate D-glucitol and L-lysine and inability to assimilate sucrose and melezitose. Thus, it should be proposed as a new species in *Tuenia*.

Typus: China, Delingha City, Qinghai province, obtained from plant tissue of *Nitraria tangutorum*, April 2021, X.L. Wei, holotype CGMCC 2.6797 preserved in a metabolically inactive state. Ex-type culture has been deposited in the Japan Collection of Microorganisms (JCM) as JCM 35,462 (=17-P246-2-1).

***Vishniacozyma pseudodimennae*** X.Z. Liu, F.Y Bai, and X.Y. Wei, **sp. nov.** (Figure 5F)

MycoBank: MB 842987

Etymology: the specific epithet ‘*pseudodimennae*’ refers to the cell morphology and colony phenotype similar to its closely related species, *V. dimennae*.

Culture characteristics: After growth on YM agar for 7 days at 20 °C, cells were subglobose to ovoid, 2.6–5.6 μm × 2.4–4.3 μm, and occurred singly or in pairs. Budding was polar. After 7 days of growth on YPD agar at 20 °C, the streak culture was whitish cream, smooth, and slightly glistening. The margin was entire. In Dalmau plated culture on corn meal agar, pseudohyphae and hyphae were not formed. Sexual structures were not observed on YCB, 5% MEA, PDA, V8, and CMA agar. Ballistoconidia were not produced on corn meal agar.

Physiological and biochemical characteristics: Glucose was not fermented. Glucose, sucrose, cellobiose, trehalose (delayed weak), lactose, raffinose, soluble starch (delayed weak), D-xylose (delayed weak), D-arabinose (weak), glycerol (delayed and weak), D-glucitol (delayed and weak), salicin (delayed), D-gluconate, DL-lactate (delayed), myo-inositol (delayed), and xylitol (delayed) were assimilated as sole carbon sources. Galactose, L-sorbose, maltose, melibiose, melezitose, inulin, L-arabinose, D-ribose, L-rhamnose, D-glucosamine, methanol, ethanol, erythritol, ribitol, galactitol, D-mannitol, Methyl-α-D-glucoside, succinate, citrate, hexadecane, and N-Acetyl-D-glucosamine were not assimilated as sole carbon sources. Ammonium sulfate, potassium nitrate, cadaverine, L-lysine, and ethylamine were assimilated as sole nitrogen sources. Sodium nitrite was not assimilated as the sole nitrogen source. Growth in the vitamin-free medium was positive (weak). Starch-like compounds were not produced. Urease activity was positive. Diazonium Blue B reaction was positive. The maximum growth temperature was 30 °C.

Physiologically, *V. pseudodimennae* differed from its close relative *V. dimennae* in the ability to assimilate potassium nitrate and inability to assimilate in galactose, L-sorbose, L-arabinose, D-ribose, L-rhamnose, ethanol, ribitol, galactitol, D-mannitol, succinate, and citrate.

Typus: China, Chaidan town, Delingha City, Qinghai province, obtained from undescribed plant tissue, April 2021, X.L. Wei, holotype CGMCC 2.6790 preserved in a metabolically inactive state. Ex-type culture has been deposited in the Japan Collection of Microorganisms (JCM) as JCM 35,455 (=25-0-5-263-4).

## 4. Discussion

The Qaidam Basin has been proposed as the first Mars analog in China as it harbors a hyper-arid, cold, salt-deposited, and high UV environment similar to the surface of Mars, as well as Mars-like geomorphological structures and mineral composition [12]. It provides an ideal systematic model as a terrestrial analog with astrobiological and extremophile implications. This study reports the culturable yeast diversity in lichen, plants, and soils in the Qaidam Basin for the first time.

Generally, yeasts grow well in a slightly acidic environment, with the optimum pH between 4.5 to 5.5 [55]. Nevertheless, basidiomycetous yeasts, i.e., *Naganishia albida*, *Papiliotrema laurentii*, *Rhodotorula glutinis*, *R*. *mucilaginosa*, *Sporobolomyces roseus,* and *Trichosporon asahii*, were found to thrive at pH value above 10 [56,57]. In our study, the pH values among the sampling sites varied from 7.54 to 9.74, indicating the alkali nature of the sampling area. Yeasts were obtained in the sampling sites with pH values between 7.65 to 8.85 but not for the soil samples with pH values greater than 9. The salinity of soil samples in our study ranged from 0.16‰ to 58.51‰. Yeasts were detected in ten soil samples with salinity from 0.21‰ and 34.32‰. Meanwhile, no yeasts occurred in seven soil samples with salinity between 0.16‰ to 16.78‰, including two soil crust samples. Though pH and salinity limited the presence of yeast inhabitants to some extent, their presence did not directly correlate with these two abiotic variables. This is probably due to additional abiotic variables affecting yeast number and diversity that have not been captured in this study.

Approximately 85% of strains obtained in our study were tremellomycetous yeast species that are in good agreement with previous studies of other extreme environments [28,29,30,34,58]. Of 21 species identified in this study, six species, such as *Diutina catenulata*, *Filobasidium magnum*, *Naganishia adeliensis*, *N. albida*, *Papiliotrema laurentii,* and *Vishniacozyma victoriae*, had been reported to be resident in extreme environments, especially saline, alkali, cold or dry conditions [59]. The most frequent yeast species were assigned to *Naganishia* in our study in that about 39.2% of isolates were identified as *N. adeliensis* and *N. albida*. The high abundance combining frequent occurrence in more than half of the samples of these two species suggested that they are resident species in the sampling area of our study [60]. *Naganishia adeliensis* and *N. albida* are versatile extremophilic species that are frequently found in acidic, alkaline, hypersaline, cold, and arid environments [59]. These two species are found as the dominant species in hypersaline soils surrounding Urmia Lake and the dry sandy soils of the desert [30,57,61]. They are characterized as capsulated yeasts that might contribute to water retention and enhance water availability [62]. *Filobasidium magnum* was also detected in our study at a higher abundance and frequency (Appendix A). *Filobasidium magnum*, which has been reported from Arctic glaciers [34,63] and plants inhabiting the Negev Desert and the Dead Sea region [64], was observed in our sampling sites in the Qaidam Basin that is characterized by hyper-arid, alkaline, and hypersaline environments. Similarly, strains of *F. magnum* were also plant-related with frequent isolation from plants in our study, implying this species may help plants to survive in dry areas. This was confirmed by the study where the application of *F. magnum* to the seed of *Brassica alboglabra* enhanced its metal resistance and its ability to grow in multi-metal contaminated soil [65]. We also isolated 32 strains of a novel species, *Filobasidium chaidanensis* sp. nov., from plants, which may own similar ecological functions as *F. magnum*. Though the other three species, *Diutina catenulata*, *Papiliotrema laurentii*, and *Vishniacozyma victoriae*, were detected in our study with low abundance and frequency, they were most commonly reported in previous studies. *P. laurentii* and *V. victoriae* displayed polyextremophilic capability, including acidophilic, alkali-tolerant, halophilic, xerotolerant, and psychrophilic aptitudes [59]. *Diutina catenulata* is a human pathogen that has been one of the most commonly reported in beach sand and estuarine and sea waters that represent alkali and saline environments [66,67].

Our study also yielded yeast species, namely *Dioszegia hungarica*, *Kondoa sorbi*, *Pichia kudriavzevii*, *Teunia korlaensis*, *Udeniomyces puniceus*, *Vishniacozyma dimennae*, and *V. tephrensis*, that have not been reported in microbial diversity research of extreme environments. Meanwhile, black yeasts, such as *Hortaea werneckii*, *Phaeotheca triangularis*, *Trimmatostroma salinum*, and *Wallemia ichthyophaga*, which are the common representatives in the hypersaline environment [59], were not found in our study. Additionally, seven undescribed species, namely *Fibulobasidium* sp., *Kondoa globosum* sp. nov., *Kondoa* sp., *Symmetrospora salmoneus* sp. nov., *Teunia nitrariae* sp. nov., *Teunia* sp., and *Vishniacozyma pseudodimennae* sp. nov., were found for the first time. Considering their low abundance and rare occurrence, they could be classified as transient species [60]. More impressively, most of these transient species colonized in plants, giving us an apparent indication of their role in plant growth in dry and alkaline soils. On the contrary, plants may help these organisms to survive in harsh conditions by transporting nutrition from soils. With this close association with plants, the transient species may obtain an intermedia to stabilize their colonization in harsh environments and then may evolve as a resident species with high abundance. The yeast diversity was significantly higher in the presence of vegetation in our study than that in the absence of vegetation, which has been proven in the Mars Desert Research Station in Utah, USA [17], probably indicating the important role of plants in the microbial growth and colonization.

Based on the delimitation of halophilic species that yeast strains can survive at salt concentration > 1%, such as seawater [68], the strains from the soil with salinities of 12.88‰ and 34.32‰ in our study have the potential to be halophilic species. Because they were distributed in lichen, plant, and soils with lower salinities, it is intriguing to know the extent of their ability to adapt to salinity variations, which will unravel the phenotypic scales of yeasts under extreme conditions. Along with the increasing NaCl concentration, the number of species that can grow well decreased from 17 species at 9% NaCl concentration to six species at 12% NaCl concentration and zero at 15% NaCl concentration. The four species that can grow at 12% NaCl concentration in our study, i.e., *F. magnum*, *N. adeliensis*, *P. laurenti*, and *V. victoriae,* have also been frequently found in the salty environment, and some of them from hypersaline lakes have been proved to tolerate 10% to 15% NaCl [30,59]. The intraspecies salt tolerance showed different tendencies. On the one hand, all the tested strains from lichen, plants, and soils have the same salt tolerance ability, such as *F. magnum*, *P. laurenti*, and *V. victoriae*. On the other hand, the adaption to salt stress should be variable inside the species. The strains of *N. adeliensis* and *S. salmoneus* from soils have extensive flexibility in changing environments compared to the strains from plants and displayed superior adaption along with elevated salinity. Previous studies indicated that halotolerant yeasts developed a number of structural and metabolic strategies to cope with the harsh environmental condition, including producing resting spores such as ascospores, teliospores, and chlamydospores [59]; adjusting the composition of a phospholipid, unsaturated fatty acid, and sterol of the yeast plasma membrane [69,70]; secreting extracellular cold-related enzymes [71] or redox enzymes to reduce the acidity [72]; synthesizing polysaccharidic capsule to prevent from drying [73]. The dynamic expression patterns of salt-dependent genes, such as ENA P-type ATPases involved in the Na^+^/K^+^ transport, Glycerol-3-phosphate dehydrogenase regulating glycerol synthesis, and *BAT2*, *ARG4,* or *AGP2* involved in amino acid metabolites, has been well studied, which provide genomic-level knowledge on the mechanisms involved in the halotolerant behaviors of yeasts, especially black yeast [69,74,75]. The underlying molecular mechanisms of halotolerance in species other than black yeasts deserve further investigation. It will increase our understanding of microorganisms in the saline environment and identify potential targets of economically important yeasts. To sum up, the biodiversity in Qaidam Basin not only gives us knowledge of life limits but also reserves resources such as halotolerant yeasts that can be used in agriculture as an eco-friendly biofertilizer to elevate crop productivity under saline conditions.

## 5. Conclusions

This is the first study that reports on the culturable yeast diversity in Delingha city in the northeast Qaidam Basin in Qinghai province, considered to be a mars analog area in China. It contributes to the general knowledge of yeast diversity in Mars analog. A total of 194 yeast isolates assigned to 12 genera and 21 species were found from ten soil samples and six plant and lichen samples with the dominance of basidiomycetous yeasts, which reveals the underestimated importance of basidiomycetous yeasts as inhabitants of the harsh environment of the Qaidam Basin desert. Furthermore, five novel species are described in this study. A significant difference in the number of yeast strains and species from plants (including one lichen sample) and soils was observed, indicating the close association between plants and yeasts. Microorganisms can be beneficial for the growth of the plant. Representative strains from different substrates of each species found in this study were subject to salt-tolerant ability from 0% to 15% NaCl concentrations. Seventeen and five species could grow at 9% and 12% NaCl concentration, while no strains could grow on the plate with 15% NaCl concentrations. Strains from soils can adapt well to the higher salt stress compared to those from plants and lichen. More comprehensive studies on yeast distribution across a wider range of sampling sites would be clearly desirable to better comprehend the distribution, ecology, diversity, and functions of yeast in the Qaidam Basin.

## Figures and Tables

**Figure 1 jof-08-00858-f001:**
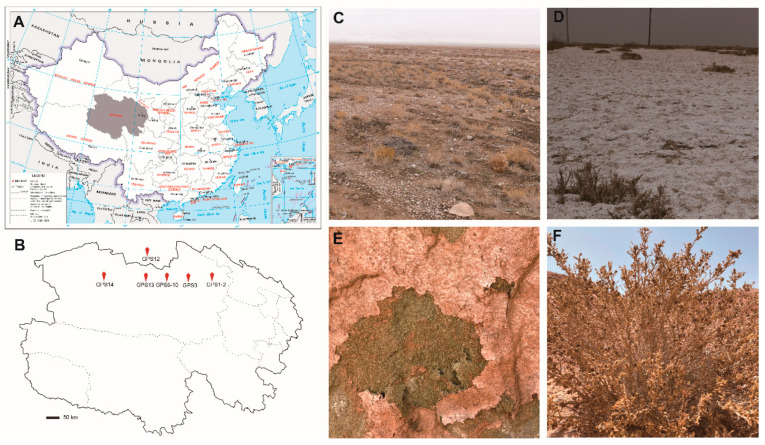
The locations and types of the sampling sites in the Qaidam Basin. (**A**) The sampling area (Qinghai province) in China, shown in grey color; (**B**) The locations of the sampling sites in Delingha city in Qinghai province labeled as GSP1–GPS3, GPS6–GPS10, and GPS12–GPS14; (**C**) Gobi desert at site GPS2; (**D**) Saline-alkali land near the Keluke lake at site GPS3; (**E**) Green sandy soil at site GPS8; (**F**) Typical plant at site GPS10.

**Figure 2 jof-08-00858-f002:**
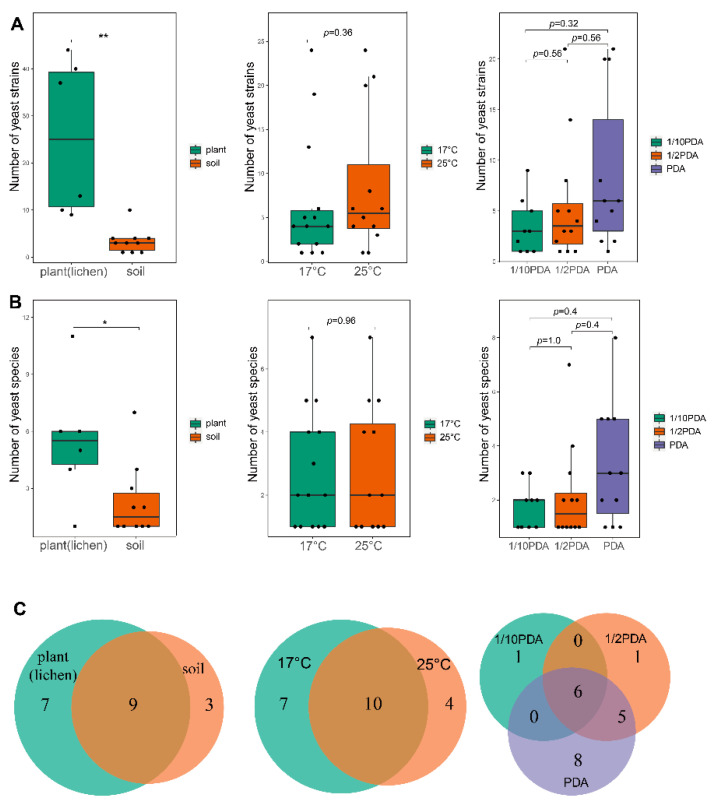
Comparison of abundance of yeast strains (**A**), species (**B**), and unique and shared yeast species (**C**) isolated from different types of samples (plant/lichen and soil) at different temperatures (17 °C and 25 °C) and using different media (PDA, 1/2 PDA, and 1/10 PDA). Single asterisk (*) represents *p* < 0.05 and double asterisk (**) represents *p* < 0.01.

**Figure 3 jof-08-00858-f003:**
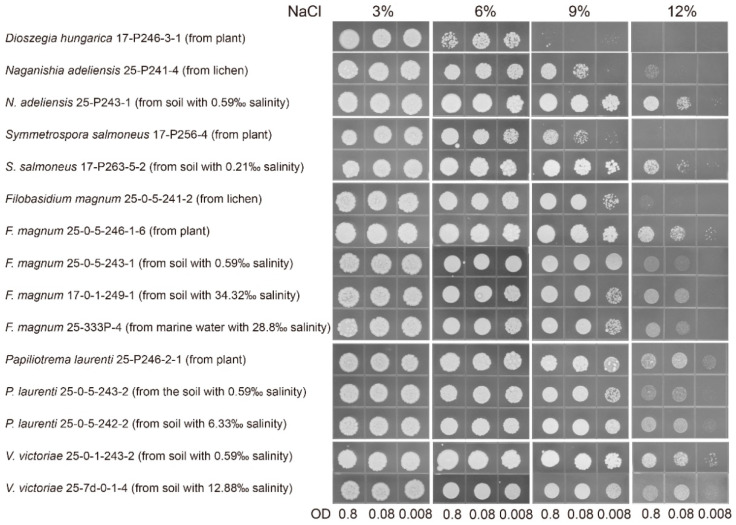
Spot plate assay for growth of representative strains of selected species on YPD medium supplemented with 0%, 3%, 6%, 9%, and 12% (*w*/*v*) NaCl for 5 days at 20 °C.

**Figure 4 jof-08-00858-f004:**
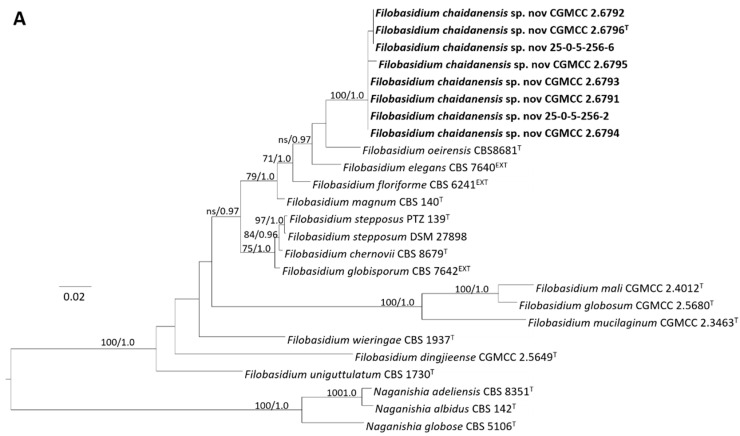
Phylogenetic trees constructed based on the concatenated sequences of the ITS and the LSU rDNA D1/D2 domains, depicting the positions of the new yeast species in the genera *Filobasidium* (**A**), *Kondoa* (**B**), *Symmetrospora* (**C**), *Teunia* (**D**), *Vishniacozyma* (**E**). The tree backbones were constructed using RAxML, and the numbers at each node represent bootstrap percentages (BP) of maximum likelihood from 1000 replicates and Bayesian posterior probabilities (PP). BP values ≥ 70 and PP values ≥ 0.9 are plotted on the branches of the tree. The new species are shown in bold. Scale in 0.02 substitution per nucleotide position.

**Figure 5 jof-08-00858-f005:**
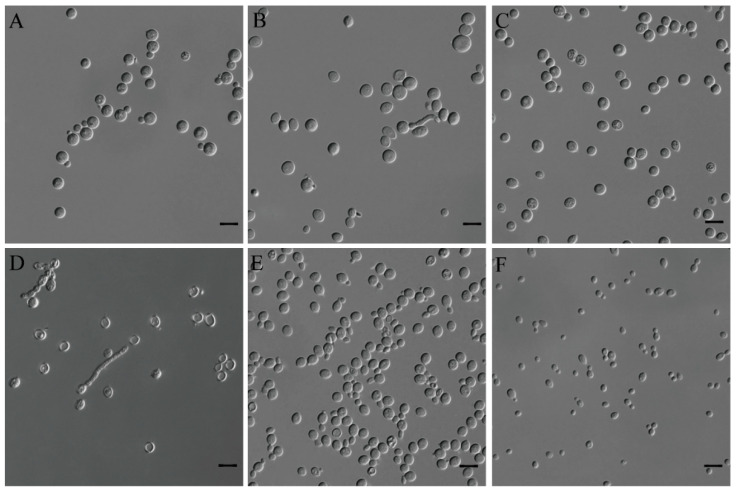
The Vegetative cells grown on YM agar for 7 days at 20 °C. (**A**) *Filobasidium chaidanensis* sp. nov. CGMCC 2.6796^T^; (**B**) *Kondoa globosum* sp. nov. CGMCC 2.6805^T^; (**C**) *Symmetrospora salmoneus* sp. nov. CGMCC 2.6801^T^; (**D**) *Symmetrospora salmoneus* sp. nov. CGMCC 2.6801^T^ × CGMCC 2.6800; (**E**) *Teunia nitrariae* sp. nov. CGMCC 2.6797^T^; (**F**) *Vishniacozyma pseudodimennae* sp. nov. CGMCC 2.6790^T^.

**Table 1 jof-08-00858-t001:** Geographical locations and chemical properties (mean ± SD) (n = 5) of soil samples collected.

Sampling Sites	Coordinates	Altitude (m)	Nature of Sample	Sample Type	Sample No.	Salinity (‰)	pH	Presence of Yeasts
GPS1	97°13′00″ E 37°20′27″ N	2922	Gobi desert	Soil	248	1.15 ± 0.00	7.93 ± 0.01	No
GPS2	97°08′44″ E 37°23′02″ N	3041.5	Gobi desert, colonized by Nitraria tangutorum and lichen	Soil	245	0.16 ± 0.00	7.86 ± 0.01	No
				Vegetation soil	247	0.25 ± 0.00	8.10 ± 0.01	Yes
				Plant	246	/	/	Yes
				Lichen(Endocarpon)	241	/	/	Yes
GPS3	96°51′15″ E 37°15′49″ N	2813.6	Saline-alkali land	Soil	249	34.32 ± 1.13	8.22 ± 0.00	Yes
				Saline-alkali soil	250	50.36 ± 0.39	8.03 ± 0.01	No
				Plant	251	/	/	Yes
GPS6	95°10′49″ E 37°34′51″ N	3499	Gobi desert	Plant	255	/	/	Yes
GPS7	95°09′41″ E 37°33′19″ N	3370	Yardang/Danxia landforms	Soil	254	1.85 ± 0.01	8.85 ± 0.01	Yes
				Salt crystal	252	58.51 ± 0.55	7.75 ± 0.01	No
GPS8	95°08′54″ E 37°31′09″ N	3261	Yardang/Danxia landforms	Red soil	263-8	1.23 ± 0.00	8.37 ± 0.01	Yes
				Sandy soil	263-5	0.21 ± 0.01	7.95 ± 0.01	Yes
				Green sandy soil	263-1	4.17 ± 0.02	7.82 ± 0.01	Yes
				Green soil crust	263-4	8.69 ± 0.02	7.54 ± 0.01	No
				Yellow soil crust	263-10	16.78 ± 0.00	7.88 ± 0.00	No
				Plant	263	/	/	Yes
GPS9	95°11′21″ E 37°32′25″ N	3354	Yardang/Danxia landforms	Soil	261	1.84 ± 0.01	8.03 ± 0.00	No
				Rhizosphere soil	260	0.69 ± 0.00	8.34 ± 0.02	Yes
				Plant	259	/	/	No
GPS10	95°12′20″ E 37°31′54″ N	3424	Yardang/Danxia landforms	Soil	258	0.26 ± 0.00	9.74 ± 0.00	No
				Rhizosphere soil	257	1.32 ± 0.01	9.01 ± 0.00	No
				Plant	256	/	/	Yes
GPS12	94°39′13″ E 38°1′36″ N	2921	Gobi desert, colonized by Nitraria tangutorum	Soil	243	0.59 ± 0.00	7.99 ± 0.01	Yes
				Plant	244	/	/	No
GPS13	94°17′23″ E 37°57′35″ N	2754	Yardang landforms	Saline-alkali soil	264	12.88 ± 0.01	7.65 ± 0.01	Yes
GPS14	93°46′19″ E 37°37′27″ N	3198	Sandy area and Saline-alkali land	Soil	242	6.33 ± 0.02	7.87 ± 0.01	Yes

## Data Availability

Publicly available datasets were analyzed in this study. All resulting alignments have been deposited in TreeBASE (http://www.treebase.org/ (accessed on 1 March 2022), accession number 29479). All newly generated sequences have been deposited in GenBank (https://www.ncbi.nlm.nih.gov/genbank/ (accessed on 28 January 2022), Appendix A). All new taxa have been deposited in Mycobank (https://www.mycobank.org/ (accessed on 13 February 2022)).

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
