# Peer review of "Yeast Diversity in the Qaidam Basin Desert in China with the Description of Five New Yeast Species"

_jof, 2022, doi:10.3390/jof8080858_

Round 1

Reviewer 1 Report

The authors have significantly improved the manuscript, therefore I recommend it for publication

Reviewer 2 Report

The extensive revision of the manuscript significantly improved its value. The authors dealt with all critical issues and notes presented by the reviewer.

 Minor comments to the revised manuscript:

p.1, lines 24-26: Naganishia albida, N. adeliensis... something is missing in the sentence (please, correct the sentence)

P.6 and other relevant pages: please correct the name Duitina catenulata (not Diutina catenulate)

p.5, line 722: A total of 91 and 103 yeast strains were discovered (please use isolated) from the plates

p.6, line 868: Filobasidium sp. occupied... please use Filobasidium sp. formed.., or Filobasidium was found...

p.7, line 1046: ... the name k. globosum sp. please change k to a capital letter K

pp. 8, 11, 12; lines 1171, 1275, 1312, 1344: Glucose fermentation is not fermented: Please correct: Glucose is not fermented.

P.13, line 1369: …inability to assimilate in sucrose and Melezitose…, please use a small letter for the word melezitose, remove in, so correct to: inability to assimilate sucrose

p. 13, line 1348: Yeasts can be detected, please correct …yeast were detected…

p. 13, line 1490: …with high frequent isolation from plants…, please remove high

p. 15, line 1492: …could enhance its metal resistance to grow in… please correct: …enhanced its metal resistance and its ability? to grow…

p. 16, lines 1599-1600: I suggest: Plants may can help these microorganisms which colonize them to survive in harsh conditions using the plant as colonization (please remove this part of sentence) and, on the contrary, microorganisms can be  good beneficial to the growth of the plant.
